# Long-Term Seasonal Drought Trends in the China-Pakistan Economic Corridor

**Sherly Shelton * and Ross D. Dixon**

Department of Earth and Atmospheric Sciences, University of Nebraska-Lincoln, Lincoln, NE 68588, USA
* Correspondence: sranasinghedisanay2@huskers.unl.edu

**Abstract:** In recent years, drought events have influenced agriculture, water-dependent industries, and energy supply in many parts of the world. The China–Pakistan Economic Corridor (CPEC) is particularly susceptible to drought events due to large-scale monsoon circulation anomalies. Using the $0.5 \times 0.5$ resolution rainfall and potential evapotranspiration data set from the Climatic Research Unit (CRU), we assessed the changes in seasonal drought variation and effects of climate variables on drought over the CPEC for the period of 1980 to 2018 using the Standardized Precipitation Evapotranspiration Index (SPEI). Our results show a statistically significant negative trend of SPEI over the hyper-arid region for two monsoons (December–February and June–September) and intra-monsoonal seasons (March–May and October–November), suggesting that the hyper-arid region (southern and southwestern part of the CPEC) is experiencing more frequent drought. A high probability for the occurrence of winter (30–35%) and summer (20–25%) droughts are observed in hyper-arid regions and gradually decreases from south to north of the CPEC. Decreasing seasonal rainfall and increasing potential evapotranspiration with increasing temperature in hyper-arid and arid regions resulted in frequent drought events during the winter monsoon season (from December to February). The findings from this study provide a theoretical basis for the drought management of the CPEC and a framework for understanding changes in drought in this region from climate projections.

**Keywords:** seasonal drought; SPEI; seasonal rainfall; trend; CPEC

## 1. Introduction

Drought, defined as when water resources are inadequate to meet the demand of people or the environment, is considered one of the most complex natural disasters due to its slow-creeping nature and nonstructural impacts [1]. Model projections of the future climate suggest an increase in drought and drought characteristics in land areas [2], especially over Central America, East Asia, the Amazon, Western Africa, and the Mediterranean [3]. The increasing frequency, severity, and intensity of drought has serious impacts on irrigation, agriculture, and hydropower, hindering the sustainable development of society [4]. For instance, the World Health Organization (WHO) estimated 55 million people who live in drought-vulnerable areas are affected by droughts every year. Climate projections also suggest increases in drought frequency and severity in Asia, Europe, the Americas, and Africa [1,5,6], and the WHO has also highlighted that 700 million people face a high risk of being displaced due to drought by 2030. Therefore, understanding the present and future drought risk in surface and groundwater, irrigation, agriculture, and socio-economic development over different parts of the world is essential for combating food and water insecurity.

South Asia has been among the perennially drought-prone regions of the world, where more than 1.5 billion people have experienced prolonged droughts [7] with high frequency and severity [8]. One of the most noticeable features of the southern Asian climate is typical of monsoon climates. The year-to-year variation of the monsoon rainfall associated with large-scale atmospheric and oceanic variability is a key driver for drought occurrence [9,10].

At the same time, high heat stress and increased air temperature can intensify the drought severity and intensity over the region [11,12] through high evaporation [13,14]. As a result of frequent drought events, drought-induced water scarcity has disrupted agricultural activities in South Asia, resulting in famines in the 19th and 20th centuries [15,16]. In the South Asian region, Pakistan is one of the most vulnerable countries to droughts strongly linked to water scarcity, food security, and the agro-economics of the country [17,18]. The climate risk index (CRI) ranked Pakistan as 5th most-affected country to climate change from 1999 to 2018 [19].

To understand the frequent drought in Pakistan, previous studies investigated the drought variability, trend, and associated mechanisms using different drought indices such as the Standard Precipitation Index (SPI) [17,20], the Reconnaissance Drought Index (RDI) [21], and the Standardized Precipitation Evapotranspiration Index (SPEI) [22], while most studies focus on drought assessment and monitoring in two cropping seasons (Rabi-November to April; Kharif-May to October). For instance, Ahmed et al. [18] found increasing drought intensity over dry and semi-arid regions for cropping seasons. Saadia et al. [17] investigated the precursor conditions that might be engaged for predicting droughts in Pakistan using an SPI. The occurrence of drought in Pakistan is closely related to anomalous total precipitation, near-surface air temperature, large-scale patterns of geopotential height, and prevailing soil moisture conditions [23]. These studies highlight the impact of hydroclimatic disasters (droughts and floods) on socio-economic, agricultural, and environmental conditions in Pakistan. Most of the climate studies in Pakistan focused on variability and trends of rainfall [24–26] and temperature [26–28].

To improve the social and economic well-being of people in underdeveloped regions of Pakistan, agriculture, health, education, energy, transport, and economic development programs are implemented [29,30] under the China–Pakistan Economic Corridor (CPEC). The CPEC is the pilot project of the Belt and Road Initiative (BRI) initiated in 2015 [31] as a collaboration between the Chinese and Pakistan governments [32]. The CPEC region is sensitive to climate change, desertification, salinization, drought, and human activity [33,34]. For instance, the CPEC is vulnerable to droughts as a result of seasonal changes in rainfall over the region, leading to food security [24]. Du et al. [35] found increasing frequency and strength of extreme precipitation, which caused floods in the CPEC region and projected increasing flood risk in the next 30 years. In addition, the CPEC has experienced dramatic changes in ecological vulnerability as a consequence of climate change and human activity [33,36]. In this context, the project shows some challenges to fully accomplishing the project objectives. Among these constraints, seasonal drought and flood hinder the project significantly.

According to the Ministry of Water Resources in Pakistan (Ministry of Water Resources, 2018), the scientific understanding of retrospective drought assessment, development of drought warning systems, and implementation of drought management plans for the CPEC is very limited [37]. The assessment of the spatio-temporal changes of the drought and flood in the CPEC is essential for adaptation to climate change, building resilient infrastructure, and for the reduction in the risks of disaster in the region. According to the author's knowledge, few studies focus on the spatial distribution of climatological seasonal drought strongly connected with circulation regimes. Therefore, the present study aims to investigate the seasonal drought trends over the CPEC routes and their relationship with temperature, rainfall, and potential evapotranspiration during 1980–2018. In this study, we used a multi-scale SPEI index that is widely adopted for drought analysis [4,38,39].

The rest of the paper is organized as follows. Section 2 presents the study site, data, and analysis methods used for our study. Section 3 focuses on presenting seasonal drought probability, the spatial trend for drought, rainfall, temperature, and PET at a seasonal time scale. Within Section 4, we compare the results with previous studies and discuss the implications and extensions of our study. Finally, in Section 5, we summarize the study and draw our conclusions on the observed seasonal drought over the CPEC.

## 2. Materials and Methods

### 2.1. Study Area

The CPEC region is located in the northwest of South Asia and extends from Kashghar in Xingjiang province (China) to Gawadar port Baluchistan Pakistan [40,41]. Within the CPEC two routes are being constructed with a total length of 4918 km. As shown in Figure 1, the eastern and western routes are located in the East and West of Pakistan, respectively. The eastern route goes across the Punjab and Sindh provinces, while most parts of the Khyber Pakhtunkhwa and Baluchistan Provinces are linked with the western route. The elevation of CPEC is complex and varies from 0 to 8611 m above mean sea level. The mountainous northern part of the CPEC is mostly comprised of glaciers and glacial lakes, while plat terrain is observed in the east part of the CPEC [29,42,43]. The CPEC receives rainfall from the Indian Summer Monsoon (ISM) and westerly disturbances initiated from the Mediterranean Sea. During the summer season, monsoon depressions originating from the Bay of Bengal bring moisture to the east and northeast parts of the CPEC during the months of July to September with high intensity (~55% to 60% of the annual budget). The CPEC shows large spatio-temporal variation in climatic conditions. The coastal climate is observed in the southern part of the CPEC, while the tropical and humid climate is dominant in the central and northern parts of the CPEC [43]. The temperature in the winter is between 4–20 °C in northern Pakistan, while in some years, it is below zero. The average temperature over the southern part of the CPEC is between 20 and 25 °C during winter. The temperatures in the north (<15 °C) and south (35 °C) parts of the CPEC show significant temperature gradients during the summer months.

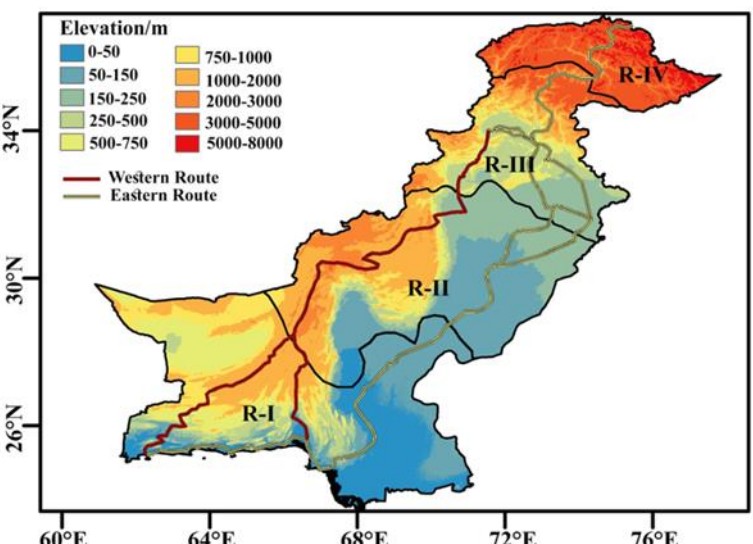

**Figure 1.** Location of the China–Pakistan Economic corridor routes and different climate regions. The black color lines demarcate hyper-arid, arid, humid, and glacial regions named R-I, R-II, R-III, and R-IV, respectively.

Based on the rainfall and temperature distribution, Rahman et al. [44] and Rahman et al. [45] have demarcated the four climate groups; hyper-arid (R-I), arid (R-II), humid (R-III), and glacial (R-IV) regions as shown in Figure 1. More than 60% of Pakistan's population resides in rural areas [37], and most of the population is concentrated in the semi-arid to hyper-arid plains of the lower Indus basin, that experience frequent drought events. In this study, we also used previously defined four climate regions for further analysis. In irrigated areas of the CPEC, wheat, maize, and rice are the main food crops, while cotton and sugarcane are considered economic crops in the region.

*2.2. Data and Methodology*

In this study, we use monthly rainfall, temperature, and potential evapotranspiration data from the Climatic Research Unit gridded Time Series (CRU TS) available at $0.5 \times 0.5$ degrees. This data set is developed by the National Centre for Atmospheric Science (NCAS) at the University of East Anglia (https://crudata.uea.ac.uk/cru/data/hrg/ accessed on 12 August 2022) and is widely used for drought analysis [46–48]. The spatial distribution of soil moisture in different seasons is plotted using CPC Soil Moisture V2 (https://psl.noaa.gov/data/gridded/data.cpcsoil.html accessed on 16 January 2023).

Calculation of Standardized Precipitation Evapotranspiration Index (SPEI)

Standardized Precipitation Index (SPI) and Standardized Precipitation Evapotranspiration Index (SPEI) [49–54] are most widely adopted for drought studies over the world because these multi-scaler indexes can be used to monitor and assess drought characteristics. Compared to the SPI, the influence of PET on drought occurrence frequency, intensity, and severity [55–57] is represented by SPEI [58]. In this study, we calculate the SPEI index for the investigation of seasonal drought variations in the CPEC. In addition, the SPI index is also used to support the validity of the SPEI results.

According to Vicente-Serrano et al. [58], the balance between precipitation (P) and potential evapotranspiration (PET) is the basic form of SPEI (Equation (1)) which can be calculated at different timescales using Equation (2) or Equation (3)

$$D = P - PET \tag{1}$$

$$D_{i,j}^{k} = \sum_{l=j-k+1}^{j} (P_{i,l} - PET_{i,l}) \text{ if } j \geq k \tag{2}$$

$$D_{i,j}^{k} = \sum_{l=13-k+j}^{12} (P_{i-1,l} - PET_{i-1,l}) + \sum_{l=1}^{j} P_{i,l} - PET_{i,l} \text{ if } j < k \tag{3}$$

where monthly potential evapotranspiration and precipitation are denoted by *PET* and *P*, respectively, while the aggregated value of monthly precipitation minus potential evapotranspiration for *k* timescale (months) is represented by *D*. The indexes *i* (year) and *j* (month) depend on the selected timescale (*k*). The log-logistic probability distribution is used to transform $D_{i,j}^{k}$, into standardized units to get multi-scalar SPEI [58].

The gridded rainfall and *PET* are then used to calculate the SPEI at 1-month (SPEI-1), to 12-month (SPEI-12) timescales. Previous studies have set "−1" SPEI as the threshold value to identify drought [59–61], and the negative SPEI values is used to determine moderate, severe, and extreme drought. In this study, SPEI-3 at February and SPEI-3 at May are selected to represent droughts in DJF and MAM seasons, respectively. For instance, as the DJF season spans from December to February, in order to consider the drought over this season, SPEI-3 February is calculated using rainfall and evapotranspiration from December to February. For the JJAS season, SPEI-4 at September is extracted from SPEI Index accumulated over four months (SPEI-4).

The nonparametric Mann-Kendall test (MK test) and Sen's slope estimator based on Kendall's tau ($\tau$) have been frequently used to analyze trends in drought [62–64]. We also used MK test and Sen's slope estimator to identify the spatial drought trend for the different seasons. We looked for a significant increase or decrease of seasonal rainfall, temperature, and PET at the 90% significance level.

## 3. Results

*3.1. Rainfall Distribution of the CPEC*

Figure 2 shows the annual cycle of rainfall distribution of different sub-regions in Pakistan. Across the entire region of Pakistan (Figure 2a) and three different climate regions (Figure 2c–e) produce bimodal rainfall distributions where the dominant peak occurs in

JJA, and another peak is observed in March. The long-term average monthly rainfall in R-I is less than 20 mm for all the months except July and August. In R-II, the average monthly rainfall is less than 35 mm, while July and August receive 50–55 mm rainfall (Figure 2b,c). R-III gets more rainfall in all the months compared to the other regions, where the maximum rainfall (140 mm) was recorded from July to August (Figure 2d). R-IV produces a different distribution than the other regions (Figure 2f), with monthly rainfall from June to November of less than 30 mm and maximum rain (50 mm) observed in March and April. The lowest rainfall for all the regions is recorded in October and November, where R-I and R-II get less than 5 mm per month while R-III gets 20 mm rainfall for this period.

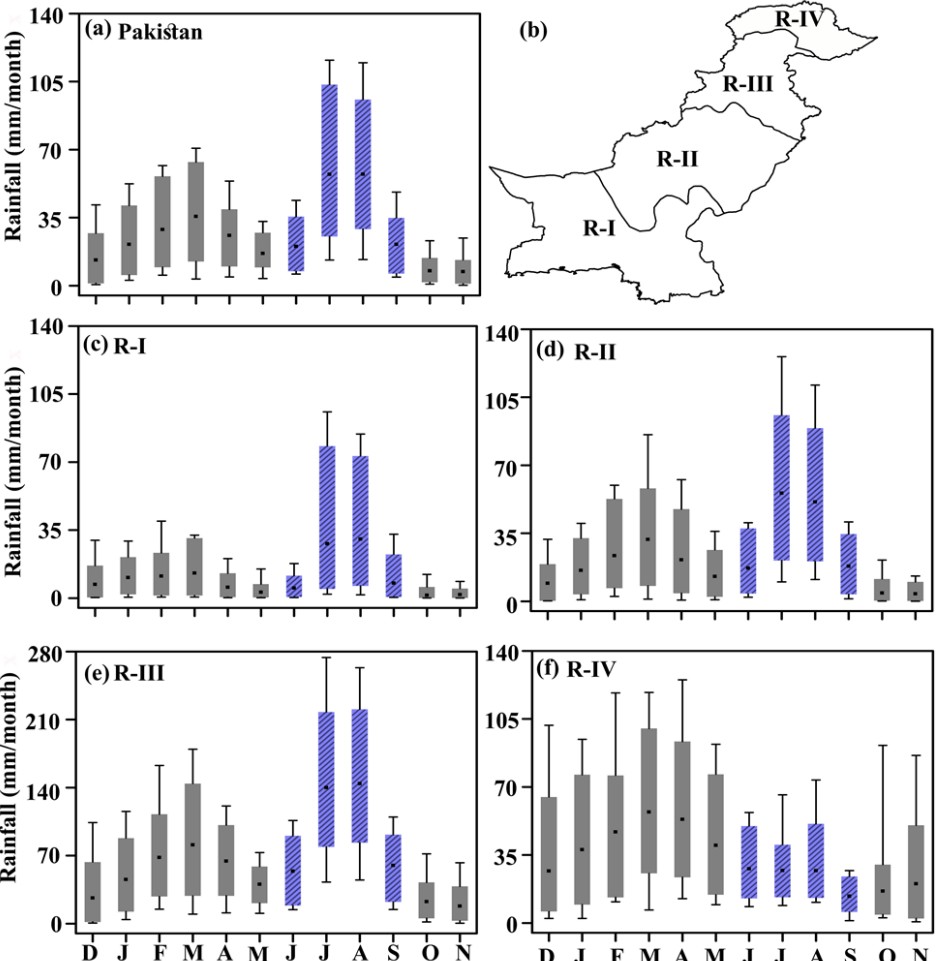

**Figure 2.** Annual cycle of the monthly total rainfall (unit: mm) in (**a**) Pakistan (**c**) hyper-arid (R-I), (**d**) arid (R-II), (**e**) humid (R-III), and (**f**) glacial (R-IV) regions for the 1980–2018 period. The blue boxes demarcate the months of the ISM season (from June to September). The long-term mean is shown by black dots, while the lines of the upper and lower whisk represent maximum and minimum rainfall, respectively. 5th (lower margin of the box), and 95th (upper margin of the box) percentiles of the rainfall are shown. The *x*-axis represents the month of the year starting from December. The four climate regions are shown in (**b**).

The findings highlight that rainfall distribution over the CPEC is closely associated with the monsoon circulation over the South Asian region. Ali et al. [25] revealed that the ISM and winter monsoon bring a significant amount of rainfall to Pakistan. July and August are the active periods of the ISM for Pakistan compared to the four months of the summer monsoon rainfall season. Wang et al. [65] found that heavy rainfall occurs in July and August in this region, and Bhatti et al. [66] observed many extreme rainfall events

in August. During the December to February period, extratropical storms initiated in the Mediterranean are responsible for the rain in the northwestern parts of Pakistan [67].

Figure 3 shows the spatial distribution of season rainfall, maximum temperature, potential evapotranspiration, and soil moisture over Pakistan for the 1980–2018 periods. During the DJF and MAM seasons, R-I and R-II received less than 50 mm per month of rainfall, while R-III gets relatively more rainfall than other regions (Figure 3a,b). The JJAS rainfall is more concentrated in the R-III, while other regions get less rainfall, especially the western part of R-I (Figure 3c). Compared to the other season, JJAS rainfall contributes to annual total rainfall in R-I, R-II, and R-III at 57.0%, 53.6%, and 52.1%, respectively; however, for R-IV, it only contributes 24% to the annual total rainfall. During the ON season, Pakistan experiences dry conditions with less rainfall (Figure 3d), contributing less than 5% in R-I, R-II, and R-III and 9.3% in R-IV. Rain from the ISM and winter monsoon seasons contributes more than 70% of the annual rainfall in all regions except R-IV.

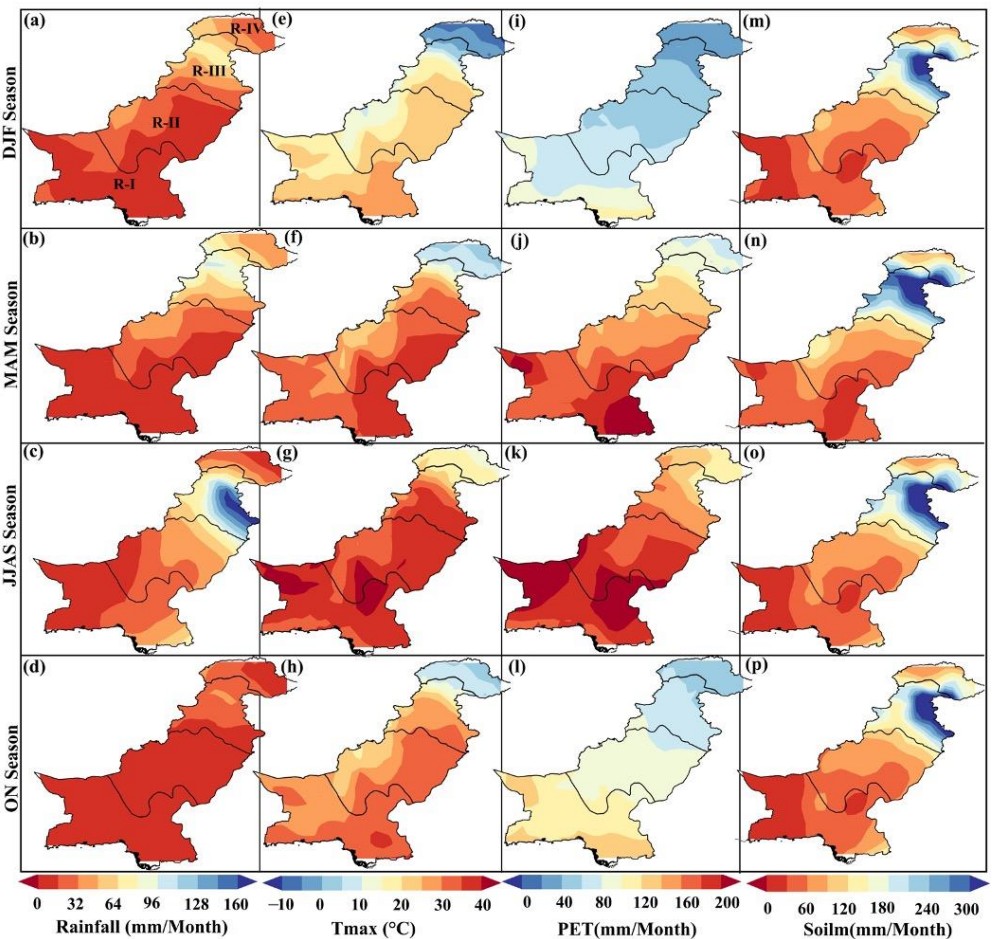

**Figure 3.** Spatial distribution of rainfall (mm/month) during the (**a**) DJF (December to February), (**b**) MAM (March to May), (**c**) JJAS (June to September), and (**d**) ON (October to November) Season over Pakistan for 1980–2018. Temperature (°C), potential evapotranspiration (PET; mm/month), and soil moisture (Soilm; mm/month) in four climate seasons are shown in (**e–h**), (**i–l**) and (**m–p**), respectively. Hyper-arid (R-I), arid (R-II), humid (R-III), and glacial (R-IV) regions are demarcated and named in (**a**).

As shown in Figure 3i–l, the lowest Tmax is observed in R-IV during the DJF season compared to other regions. R-I recorded the highest Tmax (>40 °C) in the JJAS season, followed by MAM and ON seasons. In general, the temperature in the eastern part of R-II and R-III is higher than in the western part of these two regions (Punjab and Sindh provinces). Similar to our findings, station-based observations also found that the eastern

part of the country [29] is warmer (26 to 30 °C) than the western and southwestern hilly areas (21 to 25 °C).

To determine the regional dryness and wetness and variations in meteorological conditions, PET can be used as an indicator [68,69]. Therefore, the spatial distribution of long-term PET climatology for Pakistan in different seasons is investigated (Figure 3). The lowest and highest PET is recorded in DJF and JJAS seasons, respectively. Among the regions, the lowest PET is recorded in R-IV for all seasons. Notably, the high PET is observed in R-I, and R-II occurs when Tmax is largest, suggesting that PET has a strong correlation with temperature over the region. Adnan et al. [69] used station-based PET observations and revealed that evapotranspiration in Pakistan was positively correlated with temperature, solar radiation, and wind speed. Importantly, Smakhtin and Schipper [70] found an increase in evapotranspiration under high temperatures and low humidity, which worsens the drought condition in both arid and humid regions. Based on these findings, the high temperature and low humidity may exacerbate the drought intensity and severity over R-I and R-II. Soil moisture plays a critical role in agriculture and can be used to the monitoring of the intensity of droughts [71]. As shown in Figure 3m–p, the lowest soil moisture is observed in R-I and R-II, while R-III shows the highest soil moisture irrespective of the seasons. Compared to other panels, soil moisture in the four regions is primarily associated with seasonal precipitation, while excesses in evapotranspiration (ET) due to high temperature has an additional impact, especially over the hyper-arid and arid regions.

*3.2. Drought Variation in the CPEC*

To identify the temporal variation in drought periods, a diagram showing the SPEI at different timescales (1 to 12 months) between the period 1980 and 2018 was generated for different climate regions in Pakistan (Figure 4). The cumulative effects of the temporal changes in wet and dry periods become more apparent with longer periods as the time scale of SPEI increases. As shown in Figure 4a,b, strong wet events are observed in R-I and R-II during the 1980–1998 period, which is reversed after 1999. For instance, the 1999–2003 and 2014–2018 periods exhibit continuous droughts over R-I (Figure 4a). Interestingly, we found that R-II and R-III also experienced prolonged drought events from 1999 to 2003, while the drought magnitude (−2.5 to −1.5) is much higher than R-I (−1.5 to −1.0). Similar to our findings, Ahmad et al. [72] also identified this period as the most severe drought condition in the last 50 years in Pakistan. Ahmad et al. [73] revealed that the prolonged drought from 1999 to 2003 reduced the irrigation water supply, cropped area, and crop production of Sindhi (−4% to −40%) and Baluchistan (−8% to −20%) provinces in Pakistan.

Compared to the other regions, R-IV experiences fewer drought events (Figure 3d). The findings highlight that R-I and R-II have been more susceptible to droughts in recent decades. Similar to our drought evaluation in R-I and R-II, the Indo-Gangetic plains also experienced many drought events during 2002–2004, 2009–2014, and 2015–2017 [74]. We also found that extreme wet events are dominated in the active monsoon region in Pakistan (R-III) might influence frequently occurring flood events reported over the region. Meanwhile, we were able to identify seasons with strong precipitation using these diagrams. For example, in 2010 and 2015, record-breaking monsoon rainfall was well reproduced in R-III and R-IV (Figure 4c,d), which was associated with massive flooding events (citation for flooding events). Shah et al. [75] reported severe floods in Punjab province (R-III) in 2010 with extreme rainfall in the northern parts of Pakistan. We also compared the SPEI and SPI time series in each climate region to validate the SPEI (Figure S1). The correlation between SPEI-3 and SPI-3 in all the regions is statistically significant at a 95% confidence level suggesting that the SPEI well captures the drought events in different climate regions.

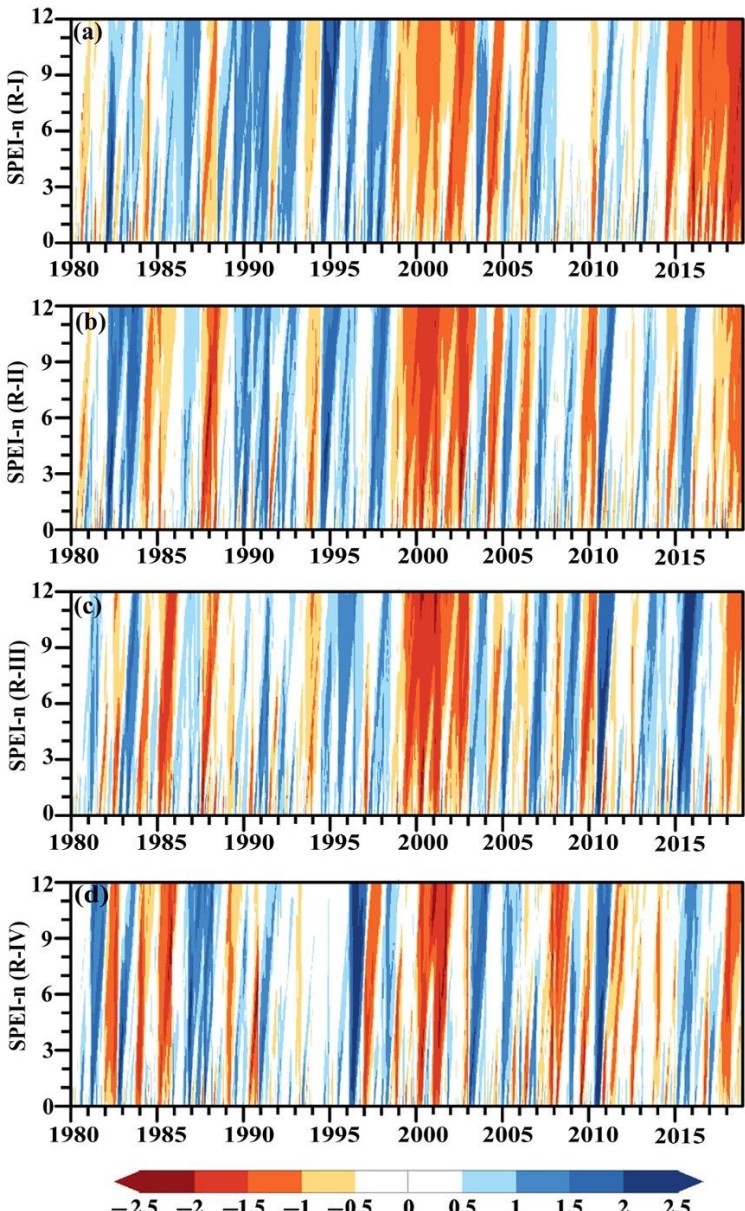

**Figure 4.** Temporal variation of the SPEI at different timescales (1 to 12 months) for (**a**) R-I, (**b**) R-II, (**c**) R-II, and (**d**) R-IV from 1985 to 2015. The R-I, R-II, R-III, and R-IV represent the hyper-arid, arid, humid, and glacial regions, respectively. The SPEI-n below −1 (or greater than 1) is selected as a threshold level for the drought (or wet) event.

*3.3. Seasonal Drought Frequency in the CPEC*

In addition to seeing how the indexes for SPEI changed for the regions, we also calculated the spatial distribution of seasonal drought frequency in Pakistan for the 1980–2018 period (Figure 5). During the DJF season, almost all of the country experiences droughts more than 15% of the time, with the strongest drought occurrence found in the southern part of the domain, especially in R-I (Figure 5a). R-I and the south part of the R-II region experience droughts more than 20% of the time during the MAM period as compared to the rest of the area (Figure 5b). During the JJAS season, drought occurrence frequency in the southwest part of the R-I region is much higher than in other regions (Figure 5c). The southern and southeast parts of R-I and R-II also experienced drought events more than 20% of the time during the ON season for the 1980–2018 period (Figure 5d). R-I experiences frequent drought events (more than 25% of the time), while drought is less frequently observed in humid and glacial regions. Similar to our findings, Jamro et al. [76]

also found frequent seasonal droughts in the arid region of Balochistan province, Pakistan. Naz et al. [77] used station data sets to calculate the SPI index and observed that the western/southwestern part of R-I and R-II has frequent droughts, which is consistent with our findings.

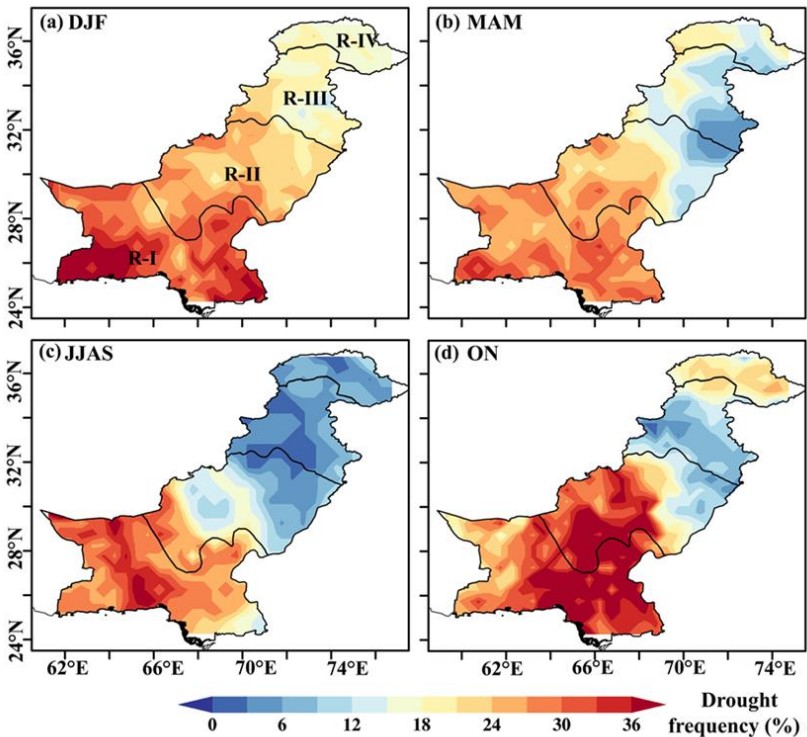

**Figure 5.** Spatial distribution of seasonal drought frequency (%) for (**a**) DJF (December to February), (**b**) MAM (March to May), (**c**) JJAS (June to September), and (**d**) ON (October to November) seasons. R-I, R-II, R-III, and R-IV represent the hyper-arid, arid, humid, and glacial regions, respectively, for the 1980–2018 periods.

### 3.4. Spatial Trend of Standardized Precipitation Evapotranspiration Index (SPEI)

While understanding the temporal variability in each region is illuminating, it is also important to consider how SPEI changes with time across the domain without being constrained by our four regions. Figure 6 shows the spatial distribution of the trend (per decade) of SPEI for the seasonal drought in Pakistan. The negative SPEI trend indicates increasing drought conditions, while the positive SPEI represents increased wetness. During the DJF and MAM seasons, a statistically significant decreasing SPEI trend is observed over R-I, suggesting that droughts are increasing from 1998 to 2018 (Figure 6a,b). The south and southwest parts of R-I, R-II, and northern parts of R-IV recorded statistically significant decreasing SPEI (Figure 6c). As shown in Figure 6d, most parts of R-I and R-II exhibit statistically significant negative SPEI, while the rest of the country has a nonsignificant negative trend for the 1980–2018 period (Figure 6d). Considering the spatial distribution of drought trends, this figure demonstrates that, for the most part, Pakistan is more vulnerable to seasonal droughts under a warming climate. This may ultimately affect crop production since Ashraf et al. [78] found that the agricultural vulnerability to drought risks increased significantly in Baluchistan (the southwest part of R-I and R-II). The calculated SPI based on station data also shows that seasonal drought is increasing in Baluchistan province [77].

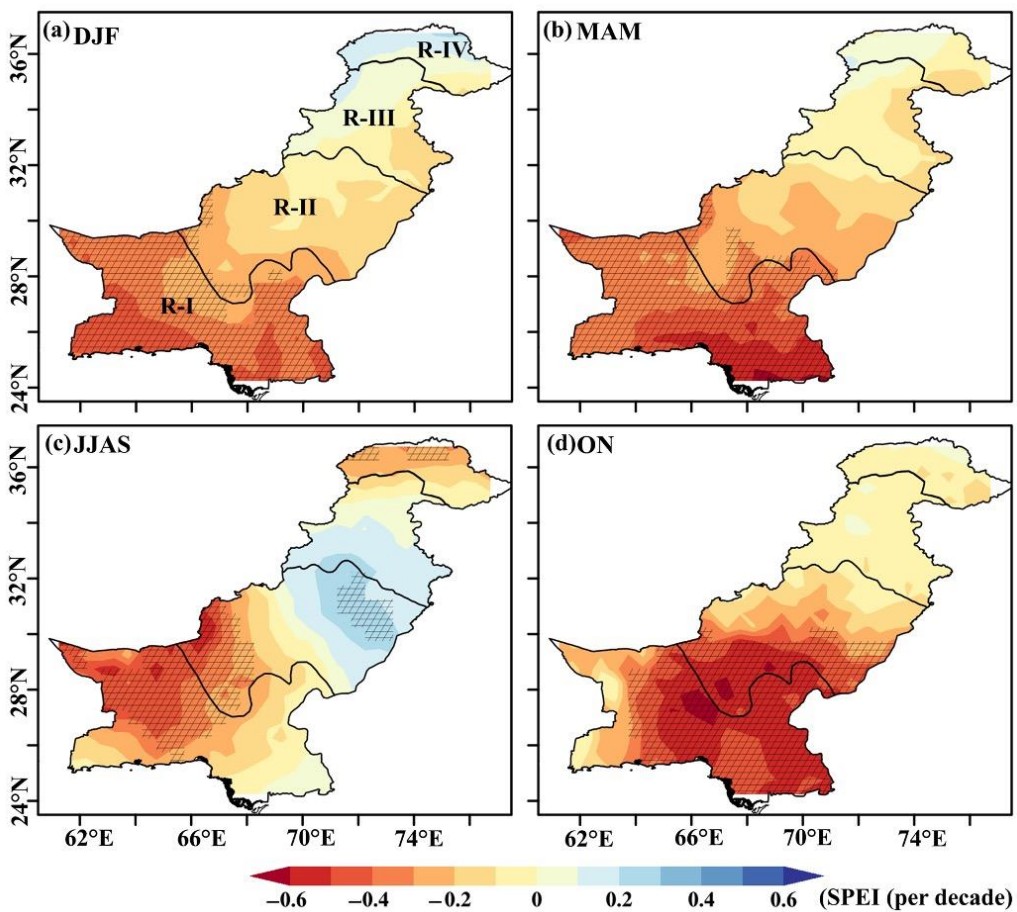

**Figure 6.** Spatial distribution of Mann–Kendal trend (per decade) of Standardized Precipitation Evapotranspiration Index (SPEI) during the (**a**) DJF (December to February; SPEI-3 at February), (**b**) MAM (March to May; SPEI-3 at May), (**c**) JJAS (June to September SPEI-4 at September), and (**d**) ON (October to November; SPEI-2 at November) Season over Pakistan for 1980–2018. The hatched area shows a statistically significant trend at a 90% confidence level. R-I, R-II, R-III, and R-IV represent the hyper-arid, arid, humid, and glacial regions, respectively.

### 3.5. Spatial Trend of Rainfall, Evapotranspiration, and Temperature

Drought occurs when there is below-average rainfall over for a protracted period, such as a few months or a season resulting in water scarcity over the region. Recent studies have found that sparse rainfall is a major reason for drought in India [74], and less rainfall coupled with increasing temperature is likely to cause seasonal drought in Balochistan, Pakistan [78]. Therefore, we investigate the seasonal rainfall trend for 1980–2018 to explain the observed drought trend over Pakistan (Figure 7). The rainfall in the DJF season shows a significant decrease over the western and southwestern parts (>3 mm per decade) of R-I and R-II, while the rest of the region (including part of R-III) depicts a statistically nonsignificant decreasing trend (Figure 7a). Ullah et al. [24] used station-based rainfall data and also found a decreasing trend of DJF rainfall in these regions.

During the MAM season, the spatial pattern of the rainfall trend is quite similar to the DJF season, but the trend magnitude is comparatively less and a small part of R-I shows a statistically significant negative rainfall trend (Figure 7b). We also identify increasing rainfall in R-III and decreasing rainfall in R-I and part of R-II during the JJAS season (Figure 7c), with notable decreases over the southwest/west part of the regions. While these are prominent trends, they are not statistically significant due to the large amount of variability from year to year in these regions (Figure S2).

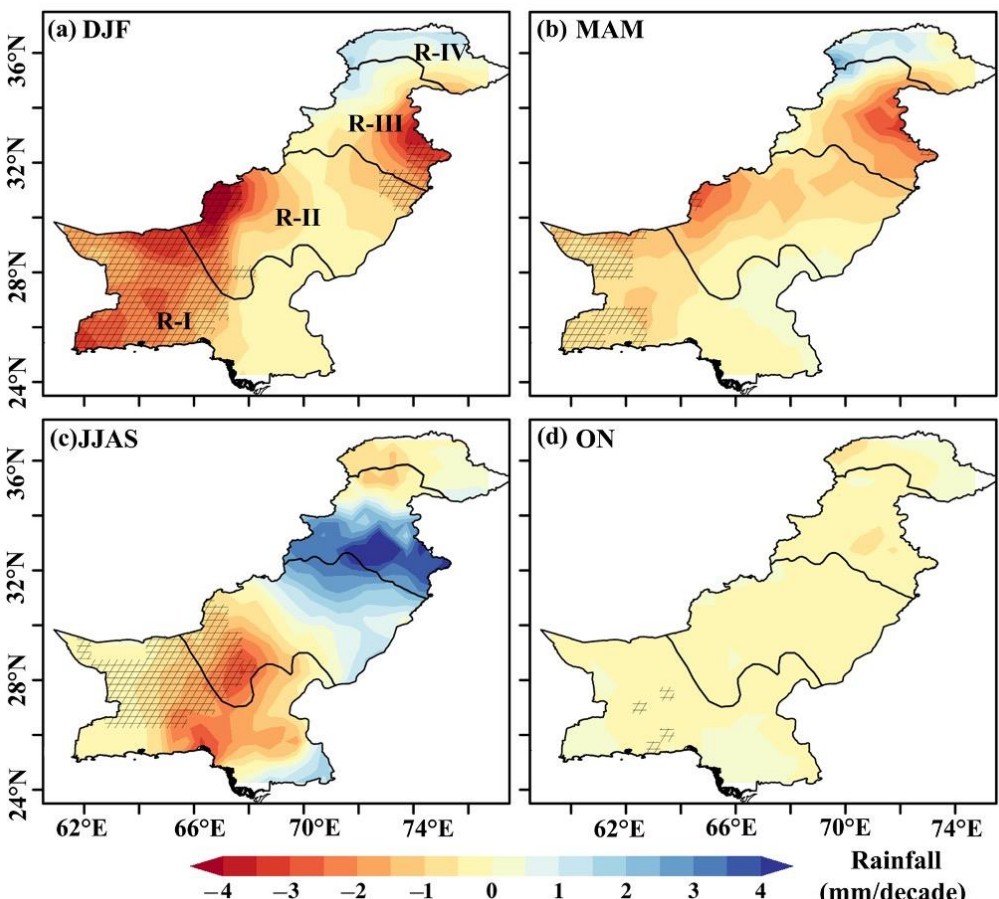

**Figure 7.** Spatial distribution of Mann–Kendal trend (mm/decade) of rainfall during the (**a**) DJF (December to February), (**b**) MAM (March to May), (**c**) JJAS (June to September), and (**d**) ON (October to November) Season over the China Pakistan Economic Corridor for 1980–2018. The hatched area shows a statistically significant trend at a 90% confidence level. R-I, R-II, R-III, and R-IV represent the hyper-arid, arid, humid, and glacial regions, respectively.

Rainfall in the ON season (Figure 7d) declined by around 1 mm per decade from 1980 to 2018. We found that very little rainfall receives during the ON season, thus a declining trend in the rainfall. Hanif et al. [79] also observed a decreasing precipitation trend in the southern part of Pakistan as well as along the coastal belt and parts of Sindh province.

The role of PET in drought is more important under global warming and changing atmospheric water demands [80,81]. Therefore, the spatial distribution of the long-term trend of PET over Pakistan is analyzed for the 1980–2018 period (Figure 8). During the two monsoon seasons (DJF and JJAS), a statistically significant increasing trend of PET is observed over R-I, while the trend magnitude in the JJAS season is higher than in the DJF season (Figure 8a,c). The increasing trend magnitude of PET for the MAM season in R-I and R-II is larger than in other seasons (Figure 8b). In the humid region, we observed statistically significant decreasing PET for all the seasons, while the largest decreasing trend is observed for the JJAS season (Figure 8c). The PET over the south and southeast part of the R-I is increasing during the ON season (Figure 8d).

In contrast, the increasing trend of PET over R-III and the upper part of the R-II are observed during the JJAS season. Ahmed et al. [82] also observed statistically significant PET trends in annual and two-crop growing seasons in recent years in Pakistan. Similar to our findings Goroshi et al. [83] found decreasing PET over monsoon regions in India.

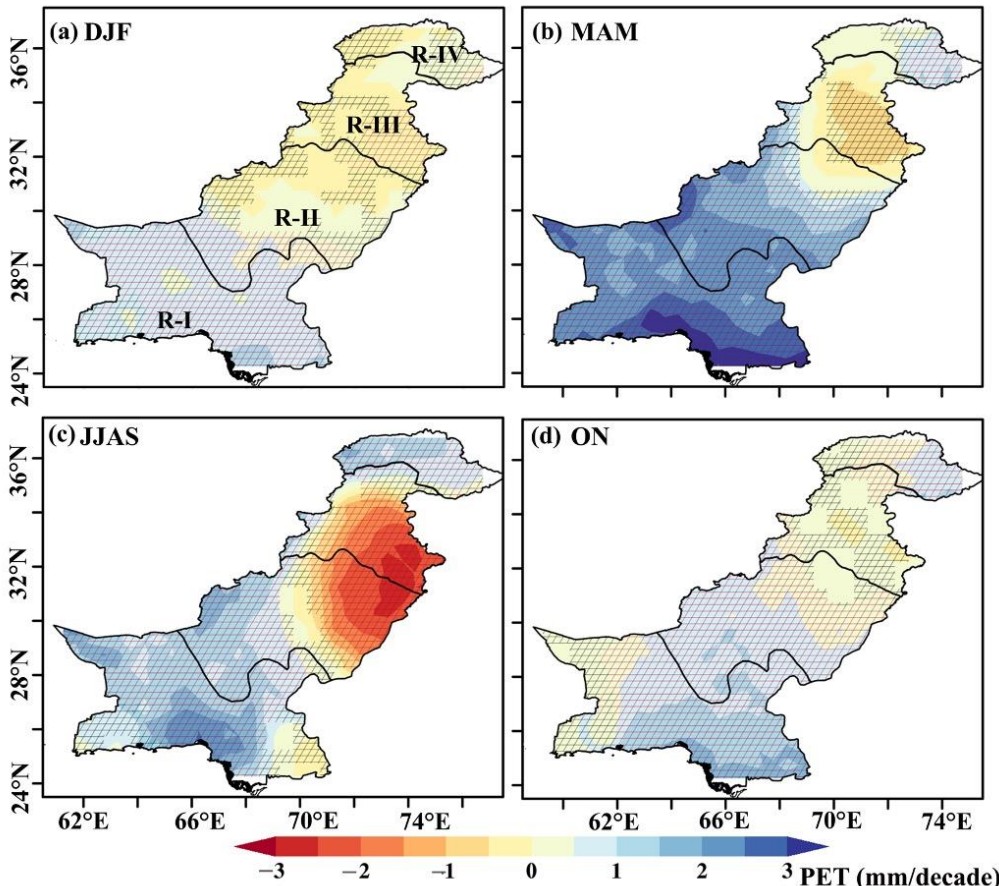

**Figure 8.** Spatial distribution of Mann–Kendal trend (mm/decade) of Potential Evapotranspiration (PET) during the (**a**) DJF (December to February), (**b**) MAM (March to May), (**c**) JJAS (June to September), and (**d**) ON (October to November) Season over the China Pakistan Economic Corridor for 1980–2018. The hatched area shows a statistically significant trend at a 90% confidence level. R-I, R-II, R-III, and R-IV represent the hyper-arid, arid, humid and glacial regions, respectively.

Increasing temperature generally leads to increased evaporative demand [84,85]. Sharma and Mujumdar [86] revealed that the combined effects of decreasing rainfall and abnormally high temperatures (heatwaves) encourage drought conditions to build and intensify than their occurrence. This study also analyzed the long-term trend of maximum and minimum temperatures in different seasons (Figure 9). The maximum and minimum temperature over Pakistan in MAM and ON seasons showed a statistically significant increasing trend where the magnitude of the trend for both Tmax and Tmin (>0.4 °C per decade) in the MAM season is higher than in other seasons (Figure 9b,f). Iqbal et al. [87] also observed the sharpest increases in Tmax and Tmin in the MAM season based on station-based observation.

During the JJAS season, Tmax and Tmin in R-I and south, southwest part of R-II showed a statistically significant increasing trend, while Tmax in R-IV also depicts a statistically significant increasing Tmax (Figure 9c). Previous studies found that Tmax and Tmin in temperature in Punjab, Pakistan increased significantly at the annual time scale [88]. Station-based observation also indicates warming in most temperature extreme indices over Pakistan [29]. In R-I and R-IV, we observed that Tmin increases faster than Tmax during all the seasons except the DJF season in R-I (Figure 9a,e). Similarly, other countries in the South Asian region have depicted an increasing trend of Tmax and Tmin where the magnitude of increasing Tmin is larger than Tmax [89,90].

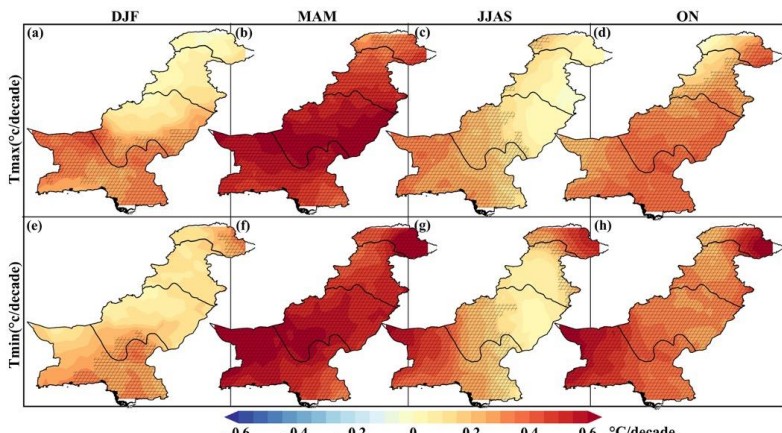

**Figure 9.** Spatial distribution of Mann–Kendal trend (°C/decade) of maximum temperature (Tmax) during the (**a**) DJF (December to February), (**b**) MAM (March to May), (**c**) JJAS (June to September), and (**d**) ON (October to November) Season over the China Pakistan Economic Corridor for 1980–2018. Lower panels (**e**–**h**) are the same as top panels but for the minimum temperature (Tmin). The hatched area shows a statistically significant trend at a 90% confidence level. R-I, R-II, R-III, and R-IV represent the hyper-arid, arid, humid, and glacial regions, respectively.

*3.6. Interactions between Precipitation, Evapotranspiration, and SPEI*

Precipitation and potential evapotranspiration are considered to calculate the SPEI in determining drought [91]. Therefore, we investigate the spatial correlation between precipitation, evapotranspiration, and SPEI, as shown in Figure 10. During the MAM, JJAS, and ON seasons, the statistically significant negative correlation between SPEI and rainfall was observed (Figure 10a–c) with a 0.9 correlation coefficient; however, the relationship between SPEI and rainfall in the winter season is comparatively less than in other seasons (Figure 10d). Yao et al. [51] revealed that rainfall was the main factor causing droughts in Xinjiang, China. The correlation between SPEI and PET over all regions is positive and statistically significant for MAM, JJAS, and ON seasons (Figure 10 a–c). The correlation between SPEI and PET is weaker (<0.4) during the DJF season (Figure 10h) and not significant for most of the domains. The findings of this study are consistent with the results of other sensitivity analyses of rainfall PET and SPEI [51,92]. However, this study shows that the relationship between SPEI, Rainfall, and PET depends on the regions and seasons.

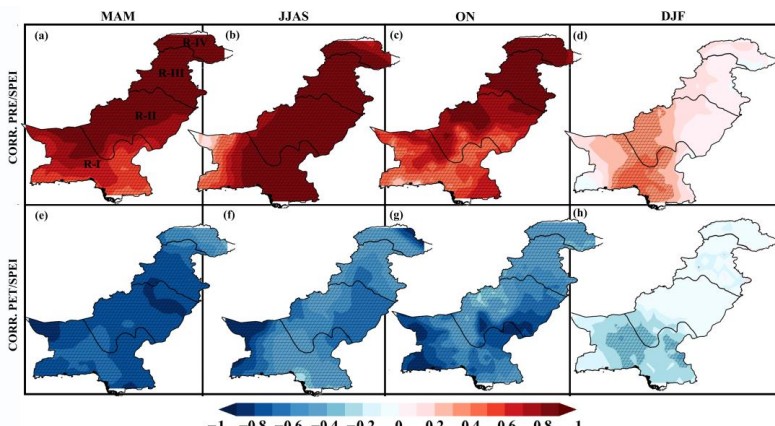

**Figure 10.** Spatial distribution of correlation between rainfall and SPEI for (**a**) MAM (March to May), (**b**) JJAS (June to September), (**c**) ON (October to November), and (**d**) DJF (December to February) seasons over the China Pakistan Economic Corridor for 1980–2018. The lower panel (**e**–**h**) is the same as the top panels but for PET. The hatched area shows a statistically significant trend at a 90% confidence level. R-I, R-II, R-III, and R-IV represent the hyper-arid, arid, humid, and glacial regions, respectively.

## 4. Discussion

Previous studies have suggested that the CPEC is highly sensitive to climate change [24,41] since most parts of the CPEC are in arid and semi-arid areas, where water resources are already scarce [41]. Frequent drought events, increasing agricultural production, and increasing population have further worsened the demand for water resources in this region [93]. To better understand the historical change in a drought, we used a CRU gridded data set and multi-scalar drought indices (SPEI) to investigate seasonal droughts in both time and space over the CPEC routes during 1980–2018. We found that across most of the southern half of Pakistan (R-I and R-II), a statistically significant decrease in SPEI was observed during the period.

When considering the water balance equation, rainfall is the supply side of the water balance that represents one aspect of drought. Actual evapotranspiration (ET) is the demand side of the water balance which depends on the moisture availability and evaporative demand (PET). Evapotranspiration is a function of temperature, humidity, solar radiation, and wind speed. With increasing temperatures, PET will increase the demand side of the water balance resulting in more severe droughts. Hyper-arid and arid regions receive less rainfall compared to other regions, and high temperatures over the region are associated with more evapotranspiration, suggesting that R-I and R-II are more susceptible to short-term and long-term droughts. Mishra and Liu [94] revealed that the risk of drought increases with fewer light precipitation days, and prolonged dry spells increase with increasing temperature. In addition, increasing the temperature intensifies the drought conditions and supports drought propagation through land-atmospheric interaction [95,96], suggesting that increasing temperature contributes to intensified drought conditions in Pakistan through an increase in evaporative demand. According to Konapala et al. [97], variations of seasonal and annual mean precipitation and evaporation influence patterns of water availability [98], crop yield [99], and ecology [100], ultimately impacting society and ecosystems. Increased PET over the R-I and R-II, in turn, enhances evapotranspiration on land and speeds up the drying of soils. Consequently, drought conditions are exacerbated and intensify droughts over the region.

In Pakistan, two cropping seasons are based on the monsoon rainfall: Rabi crop depends on the winter monsoon (DJF) rainfall, while Kharif crops are cultivated during the summer monsoon (JJAS) [101,102]. Under the warming climate, the decreasing rainfall and increasing evapotranspiration intensify plant drought stress and reduce/degrade both Rabi and Kharif vegetation in water-limited hyper-arid and arid regions. In parallel with our findings, Kharif and Rabi aridity over the southern part of the country is increased [101]. It was just anticipation based on the assumption that increased drought due to decreasing rainfall and rising temperature intensifies potential evapotranspiration (PET) over the region. As a result of water scarcity induced by frequent drought events, agricultural production will decrease in arid and semi-arid regions and end with food insecurity for 70–80% of the population in Pakistan who are engaged with the environment and weather-vulnerable farming [78].

As a response to increasing temperature, glacier retreat ultimately leads to changes in water discharge with gradually decreasing available ice reserves [103]. Kääb et al. [104] found that glaciers in the Jammu–Kashmir region (located in R-IV) are melting at the rate of 0.66 m/year under global warming. With this perspective, increasing Tmax and Tmin in the glacial region in Pakistan results in glaciers melting faster than normal, which can trigger flash flooding in the northern part of Pakistan. For instance, a glacial lake outburst flood in 2022 destroyed Hassanabad Bridge in Pakistan's northern Gilgit-Baltistan region. We found that over the humid region (R-III) an increase in summer rainfall was associated with decreased drought. Safdar et al. [105] also found an increase in monsoon-hit areas located in the northern part of Pakistan with rainfall ≥2.5 mm/day. According to the climate projection studies using CMIP6 models [106], the ISM is increasing by 0.33 mm/day and 5.3% per Kelvin of global warming. In Pakistan, monsoon rainfall is one of the major

reasons for floods, while the snowmelt intensifies flooding in rivers during the monsoon period [75].

Monsoonal variability and inconsistency can be partially attributed to large-scale circulations such as IOD and ENSO [105]. Previous studies in Pakistan revealed that rainfall variability is strongly associated with large-scale teleconnections [17,107]. Therefore, understanding the influence of the tropical sea surface temperature (SST) anomalies on observed seasonal drought variability over Pakistan may help explain the increase in intensity and frequency of droughts in recent decades. To identify the dominant precursory oceanic signals on seasonal drought characteristics, numerical simulations with a climate system model and observation-based statistical analysis will be needed. For instance, statistical analysis on reanalysis data found that circulations associated with Pacific and Indian Ocean SSTs are highly associated with drought episodes across the Pakistan region [17].

Except for meteorological forcings and their variability, droughts are initiated and maintained by several factors, such as soil and vegetation feedback, agricultural practices, and management choices, including irrigation and grazing density. Among these factors, soil moisture variability and trends are crucial to quantify the intensity of droughts [71]. In this study, drought analysis is based only on temperature and precipitation and does not make use of soil moisture content and moisture deficits in Pakistan. Therefore, future studies may wish to investigate the influence of drought on soil moisture in both the surface and root zones, as well as identify how changes in soil moisture impact the changing drought using in situ observations, integrating satellite soil moisture data or soil moisture data from hydrological or land surface models.

Though it is challenging to set up drought policies and preparedness plans, the outcomes of the process can significantly increase societal resilience to drought and drought-related disasters. Based on the findings, drought in the different climate regions depends on the climatological seasons. In addition, we identified hyper-arid and arid regions as the most drought-prone regions in the CPEC. These facts encourage the improvement and application of seasonal and shorter-term drought forecasts. Furthermore, we highlighted the priority of developing integrated monitoring and drought early warning systems and associated information delivery at regional scales, especially over the hyper-arid and arid regions. The findings of this study are also helpful in raising awareness at the regional level and developing a host of preventive and mitigation measures against droughts. In the drought policy and preparedness process, inventory data and financial resources available and identifying groups at risk are crucial. We observe that drought in the hyper-arid and arid regions is intensifying. Therefore, government and non-government organizations can prioritize identifying the people at high risks, such as agricultural and livestock communities. Establishing a drought task force is recommended to understand drought mitigation techniques, risk analyses (economic, environmental, and social aspects), and drought-related decision-making processes.

## 5. Conclusions

The CPEC is very sensitive to natural disasters such as floods and droughts under global warming. Therefore, monitoring drought and floods at seasonal timescales is a vital and timely task to mitigate the impacts of recurrent and unexpected drought and flood events. For this reason, this study investigates the seasonal drought variation and its associated trends from 1980 to 2018 over the region using a multi-scalar drought index (SPEI) which represents the effect of precipitation and PET on drought characteristics. For calculating SPEI at a seasonal time scale, long-term (1980–2018) $0.5° \times 0.5°$ resolution gridded data of monthly PET and rainfall was used. The results suggest that hyper-arid and western and southwestern parts of arid regions experience frequent seasonal droughts compared to the rest of the region in Pakistan. We also found that an increase in drought occurrence in hyper-arid and southern and southwestern parts of the arid regions was attributed to decreasing seasonal rainfall and increasing potential evapotranspiration with increasing temperature.

This study provides a scientific understanding of seasonal droughts for drought monitoring and implementing drought early warning systems at local and national levels. However, effective communication of the results at local, state, and national levels, with environmental, social, and economic perspectives, is crucial. In addition, predicting the probability of occurrence of seasonal droughts and establishing a skillful drought prediction system is essential and useful to society concerning water resource management and agricultural planning. Niranjan Kumar et al. [108] found that predicting seasonal droughts, especially during the monsoon season is challenging. Therefore, we have to improve our regional and global models' ability to simulate the drivers of hydrological change across this region and increase model skill in seasonal to decadal prediction of precipitation change. The findings of our study can be used for implementing drought adaptation strategies for future droughts expected in the coming decades. Therefore, identifying regional drought risk and improving the prediction of regional drought risk under future scenarios is crucial to provide stakeholders with useful information in order to prepare for changes in extreme regional events.

**Supplementary Materials:** The following supporting information can be downloaded at: https://www.mdpi.com/article/10.3390/cli11020045/s1, Figure S1 Time series of SPEI-3 and SPI-3 for (a) hyper-arid (b) arid (c) humid and (d) glacial regions in CPEC from 1980 to 2018. The positive and negative SPEI-3 values are shown by blue and pink colored shadings, respectively. The black line indicates an SPI-3 time series. Figure S2: Standard deviation of seasonal rainfall during (a) DJF (December to February), (b) MAM (March to May), (c) JJAS (June to September), and (d) ON (October to November) season over the China Pakistan Economic Corridor for 1980–2018; Figure S3: Spatial distribution of self-calibrating Palmer Drought Severity Index using Pen-man-Monteith (sc_PDSI_pm) during the (a) DJF (December to February; SPEI-3 at February), (b) MAM (March to May; SPEI-3 at May), (c) JJAS (June to September SPEI-4 at September), and (d) ON (October to November; SPEI-2 at November) Season over Pakistan for 1980–2018. R-I, R-II, R-III, and R-IV represent the hyper-arid, arid, humid, and glacial regions, respectively.

**Author Contributions:** Conceptualization, S.S. and R.D.D.; methodology, S.S.; formal analysis, S.S.; writing—original draft preparation, S.S.; writing—review and editing, R.D.D.; visualization, S.S.; supervision, R.D.D. All authors have read and agreed to the published version of the manuscript.

**Funding:** This research received no external funding.

**Data Availability Statement:** The gridded rainfall, temperature, and potential evapotranspiration data are freely available from the National Centre for Atmospheric Science (NCAS) at the University of East Anglia (https://crudata.uea.ac.uk/cru/data/hrg/ accessed on 12 August 2022).

**Acknowledgments:** We thank the National Centre for Atmospheric Science (NCAS) and the Climatic Research Unit (CRU) at the University of East Anglia for providing the CRU TS dataset. We thank the Editor and three anonymous reviewers for their comments which helped improve the manuscript.

**Conflicts of Interest:** The authors declare no conflict of interest.

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
