# Peer review of "Long-Term Seasonal Drought Trends in the China-Pakistan Economic Corridor"

_climate, doi:10.3390/cli11020045_

Round 1

Reviewer 1 Report

It has been a delightful experience to review the manuscript "Long-term Seasonal Drought Trends in the China-Pakistan Economic Corridor."

The work has been conducted with scientific integrity and robustness. 

I have certain queries and suggestions for the improvement of the paper

One of the main limitations of this work is the reliance on temperature and precipitation alone for the evaluation of drought index. The authors have themselves pointed out it in the last paragraph of their discussion. However, I do not see any reason why not that has been included as there are global datasets pertaining to soil moisture available for the analysis. I would strongly suggest the author to incorporate that as well. 

Second is the use of SPEI alone. The authors could have used a multi drought indexing approach. This would have provided a multi-model comparison. The issue is the validity of the SPEI results. Using a multi-indexing approach, that uncertainty could have been averted if there was any issue of non-availability of the observational data for validation.

Third, issue is the discussion regarding the policy framework involved according to the results of this research. CPEC has huge importance. How Pakistan and China are going to deal with the issues regarding drought in this area needs to be discussed. What are suggestions from this research for policymakers that need to be emphasized? 

The authors are requested to incorporate these suggestions. 

Best of luck

Reviewer 2 Report

#1 My major comment is that the writing style should be heavily improved for journal publication.

#2 In addition, the novelty/significance of the study isn’t clear. The authors focus on the monsoon season in the introduction, but in the methodology/results, they present all timescale droughts (1- to 12-month drought). The results should be well summarized to demonstrate the broader implications of this work.

#3 What is driving the droughts in arid, hyper-arid... regions should be discussed more thoroughly (and should be included in the conclusion and abstract).

Therefore I recommend resubmission of a heavily revised version.

#4 L18-19:  “a  framework for understanding changes in drought in this region from climate projections”. Unless you thoroughly discuss what is driving drought (P/PET/T etc) it is difficult to figure out the significance of this study in regional climate projections

#5 Much better writing is expected for journal publication. For example, L23-25, L31-33, L514-516

#6 L69: Review the paper [31] and demonstrate the novelty of this work in relation to the paper [31]

#7 L74-76: This sentence is not clear. What are the authors trying to say?

#8 L78-81: Why the authors suddenly introduced floods while concluding the paragraph mostly on drought?

#9 Figure 1: Aridity regions should be delineated more clearly as in Figures 3 and 5.

#10 L148: use consistent citation format

#11 Figure 4: I suggest authors analyze and plot separate figures for a timescale of drought. Fewer timescales. for example, 3 months for seasonality, and 12 months for annual/water year timescale might show a more clear picture. See comment #2.

#12 Figure 10: Colorbar scale is > |1|. correlation varies [-1,1]. Something is wrong with the calculation

 #13 L526-531: Analysis/results are based on drought but the authors concluded the paper with flood risk. weird!

Reviewer 3 Report

Dear author(s), Thank you very much for submitting your valuable contribution to Climate. I read this article few time and find very interesting. I saw most of the section has significantly improved and no need revision. However, in some lines and sentences there still minor issues.  I am requesting the authors to have a relook and correct it before final acceptance. Please also clearly define the policy recommendation in order to protect the planet. Thanks.

Round 2

Reviewer 1 Report

Accept in present form

Author Response

No comments to respons.

Indeed, we highly appreciate the anonymous reviewer-01 for his/her complimentary comments.

Thank You!

Reviewer 2 Report

The authors have responded to my review comments and improved the manuscript accordingly. 

minor comment

Since this study is on drought, don't focus on floods in the manuscript. For example, the authors introduced flood at the beginning of the conclusion (Line 539) and concluded the paper with flood risk (L565-568). I noted this in my previous review. Does the results/discussion of this study help flood mitigation/management?

Author Response

Dear Editor,

We wish to thank you for facilitating to get valuable comments from reviewers on time, which are very helpful for improving the manuscript. Indeed, we highly appreciate the anonymous reviewer-02 for his/her complimentary comments.

We have responded to each of the specific comments raised by the reviewers. A summary of the point-by-point response to the reviewer’s comments is attached in the document labeled, “Response to reviewers-02”. We hope that you find we have satisfactorily responded to the concerns raised by the reviewers in this latest set of revisions and that the manuscript is now acceptable for publication. If you have any additional questions or concerns, please do not hesitate to contact us

Sincerely,

Sherly Shelton

Reviewer- 02

  1. Since this study is on drought, don't focus on floods in the manuscript. For example, the authors introduced flood at the beginning of the conclusion (Line 539) and concluded the paper with flood risk (L565-568). I noted this in my previous review. Does the results/discussion of this study help flood mitigation/management?. 

Author's Response: We thank the reviewer for his/her detailed review and comments

We agree with the reviewer’s suggestion. Yes, we removed the sentences as you can see in lines 563-567 on page 17 (Marked with green color).